# Robust and Efficient Assessment of Potency (REAP) as a quantitative tool for dose-response curve estimation

**Shouhao Zhou[1]\*[†], Xinyi Liu[1†], Xinying Fang[1†], Vernon M Chinchilli[1], Michael Wang[2], Hong-Gang Wang[3,4], Nikolay V Dokholyan[3,5], Chan Shen[1,6], J Jack Lee[7]**

[1]Department of Public Health Sciences, Pennsylvania State University, Hershey, United States; [2]Department of Lymphoma and Myeloma, University of Texas MD Anderson Cancer Center, Houston, United States; [3]Department of Pharmacology, Pennsylvania State University, Hershey, United States; [4]Department of Pediatrics, Pennsylvania State University, Hershey, United States; [5]Department of Biochemistry and Molecular Biology, Pennsylvania State University, Hershey, United States; [6]Department of Surgery, The Pennsylvania State University, Hershey, United States; [7]Department of Biostatistics, University of Texas MD Anderson Cancer Center, Houston, United States

**\*For correspondence:**
shouhao.zhou@psu.edu

[†]These authors contributed equally to this work

**Competing interest:** The authors declare that no competing interests exist.

**Abstract** The median-effect equation has been widely used to describe the dose-response relationship and identify compounds that activate or inhibit specific disease targets in contemporary drug discovery. However, the experimental data often contain extreme responses, which may significantly impair the estimation accuracy and impede valid quantitative assessment in the standard estimation procedure. To improve the quantitative estimation of the dose-response relationship, we introduce a novel approach based on robust beta regression. Substantive simulation studies under various scenarios demonstrate solid evidence that the proposed approach consistently provides robust estimation for the median-effect equation, particularly when there are extreme outcome observations. Moreover, simulation studies illustrate that the proposed approach also provides a narrower confidence interval, suggesting a higher power in statistical testing. Finally, to efficiently and conveniently perform common lab data analyses, we develop a freely accessible web-based analytic tool to facilitate the quantitative implementation of the proposed approach for the scientific community.

## Editor's evaluation

This article proposes methodology and accompanying software for robustly fitting dose-response curves where response is a number between 0 and 1. When response is transformed using the common logistic transformation, values close to 0 or 1 become large in magnitude, unduly influencing the fitted curve after back-transformation and introducing bias in the estimate of certain parameters. As demonstrated through simulation and application to real data, the proposed approach, called Robust and Efficient Assessment of Potency, is less perturbed by these extreme measurements.

## Introduction

The median-effect equation is a unified theory in medicine to describe the dose-response relationship and identify agents or their combinations that activate or inhibit specific disease targets (**_Chou,_**

**eLife digest** Finding a new drug which is both safe and efficient is an expensive and time-consuming endeavour. In particular, establishing the 'dose-effect relationship' – how beneficial a drug is at different dosages – can be challenging. Predicting this curve requires gathering experimental data by exposing and recording how cells respond to various levels of the drug. However, extreme values are often observed at low and high dosages, potentially introducing errors that are hard to correct in the prediction process. Yet, these extreme observations are sometimes genuine so researchers cannot just ignore them.

To improve dose-effect estimation, Zhou, Liu, Fang et al. developed a new general-purpose approach. It uses advanced statistical modelling to account for extremes in lab data. This strategy outperformed other methods when dealing with these observations while also providing higher efficiency in data analysis with more uniform data in experiments.

To facilitate implementation, Zhou, Liu, Fang et al. set up a user-friendly tool baptized 'REAP'; this free online resource allows scientists without advanced statistical experience to harness the new approach and to perform dose-effect analysis more easily and accurately. This could boost research across many different disciplines that examine the effects of chemicals on cells.

2006). It is a fundamental method established based on the pharmacological principle of mass-action law (*Chou, 1976*). As the common link for many biomedical systems, it has been used extensively to analyze in vitro experimental data and evaluate the potency of related drugs (*Chou and Talalay, 1984*; *Chou and Rideout, 1991*; *Greco et al., 1995*; *Lee and Kong, 2009*).

In practice, the median-effect equation can be estimated for drug efficacy or pathway inhibition from normalized data generated from experimental studies. Without knowing the true dose-effect curve during the experimental design and data collection, it is common to observe extreme values of (un)affected cell fraction that is close to the response of either 0 or 100% in the analytic dataset. Quantitatively, it poses a special analytic challenge to estimate the median-effect question in practice. The standard estimation approach, often based on a linear regression model after a logit transformation (*Roell et al., 2017*; *Gadagkar and Call, 2015*), could suffer badly from poor estimation in such situations. *Figure 1* illustrates a preliminary example in that the standard approach is deficient in describing the median effect curve with a perturbation in one extreme data point. The variation in real experimental data, mostly caused by unavoidable measurement error, often at a much larger degree, therefore challenges the reliability of result presentation and interpretation for many drug assessment studies.

Additionally, the modeling strategy of deleting extreme values may not be feasible in many situations (*Solzin et al., 2020*). For example, a meaningful drug concentration could consist of high inhibition (>90%) or low cell viability (<10%) in cancer research. It is not logical to ignore extreme observations when they are indeed biologically relevant for the target effect, not even to mention an associated loss of power and accuracy by leaving fewer data points for estimation. As illustrated in *Figure 2*, deleting the extreme values couldn't eliminate the estimation bias, but only impaired the efficiency of interval estimation with wider nominal 95% confidence intervals (C.I.) and harmed the estimation accuracy with worse coverage probabilities.

Furthermore, it is dubious to apply the constant error variance, a default assumption in standard linear regression modeling, in dose-response estimation. As an assumption can be examined with repeated measures, many dose-response data have indicated either a constant variance before logit transformation or a positive correlation with drug dose. It is incongruous to apply linear regression if the assumption is violated due to error heteroscedasticity (*Schmidheiny, 2009*; *Williams et al., 2007*). Therefore, it is essential to develop a robust quantitative approach to estimating the median-effect equation.

Here, we introduce a novel approach to improving the quantitative assessment of dose-response relationship and drug potency, together with a user-friendly web-based analytic tool to facilitate the implementation. The proposed method to estimate the median-effect equation is established in the robust beta regression framework, which not only takes the beta law to account for non-normality and heteroskedasticity (*Ferrari and Cribari-Neto, 2004*), but also minimizes the average density power

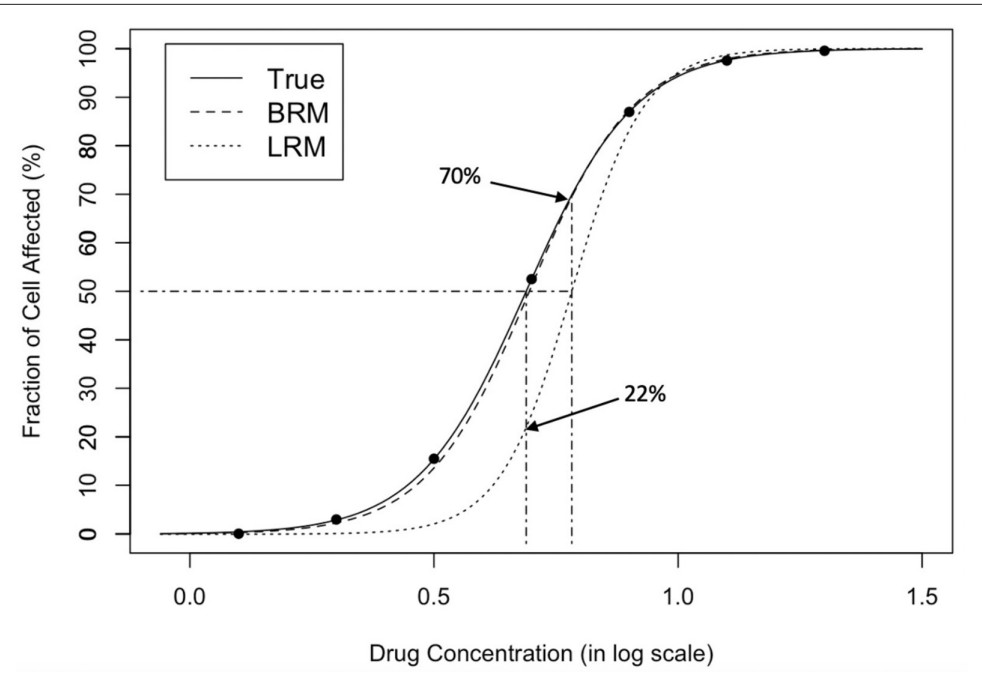

**Figure 1.** Dose-response curve fitting with extreme observations. The original data points are on the true curve. The leftmost data point is changed from 0.005 to 1e-6, referring to a small white noise that cannot be visually recognized. The change leads to the obvious departure between the estimated curve by linear regression model (dotted) and the true curve (solid), which demonstrates that standard regression is sensitive to extreme values. The response at the true $IC_{50}$ (dotdashed, vertical, left) is only 22% from the estimated curve; the estimated $IC_{50}$ (dotdashed, vertical, right) corresponds to the 70% fraction of cell affected, effecting a substantive 20% inflation (50% ->70%) in estimation error. In contrast, the estimated curve by beta regression model (dashed) is almost overlapped with the true curve (solid), which shows that BRM is much more robust to extreme values. LRM: linear regression model; BRM: robust beta regression model. Detailed model descriptions of LRM and BRM are provided in Materials and methods section.

divergence (DPD) using a tuning parameter (*Ghosh, 2019*). We apply a data-driven approach to optimizing the tuning parameter, which further compensates for the lack of robustness against outliers. In the simulation studies, we compare the robust beta regression framework with linear regression models either in the standard normal distribution error, or in the heavy-tailed t distribution error with 3 degrees of freedom hopefully to downweigh the influence of extreme observations. Results from simulation studies under various scenarios confirm that the proposed approach consistently gives robust estimation for the median-effect equation. Particularly, we examine two important measures for drug binding affinity: the Hill coefficient, which signifies the sigmoidicity of the curve, and the overall effect, indicated by dose concentration for a specified (e.g. 50%) response (*Shen et al., 2008*; *Sampah et al., 2011*). When there are extreme outcome observations, the improvement of robust beta regression in estimation accuracy could be substantial. Moreover, simulation studies further illustrate that the proposed approach provides a narrower confidence interval, which in turn suggests a higher efficiency to achieve better power in statistical testing even without acquiring additional experimental data. Illustrative examples using real-world data for cancer research and SARS-CoV-2 treatment are provided. The analyses are implemented using the freely accessible web-based application REAP, developed based on the Shiny package of R language, with which research scientists could conveniently upload their drug experiment dataset and perform the data analysis.

## Results
### REAP Shiny App
We developed a user-friendly analytic tool, coined 'REAP' (Robust and Efficient Assessment of Potency), for convenient application of the robust dose-response estimation to real-world data analysis. It is

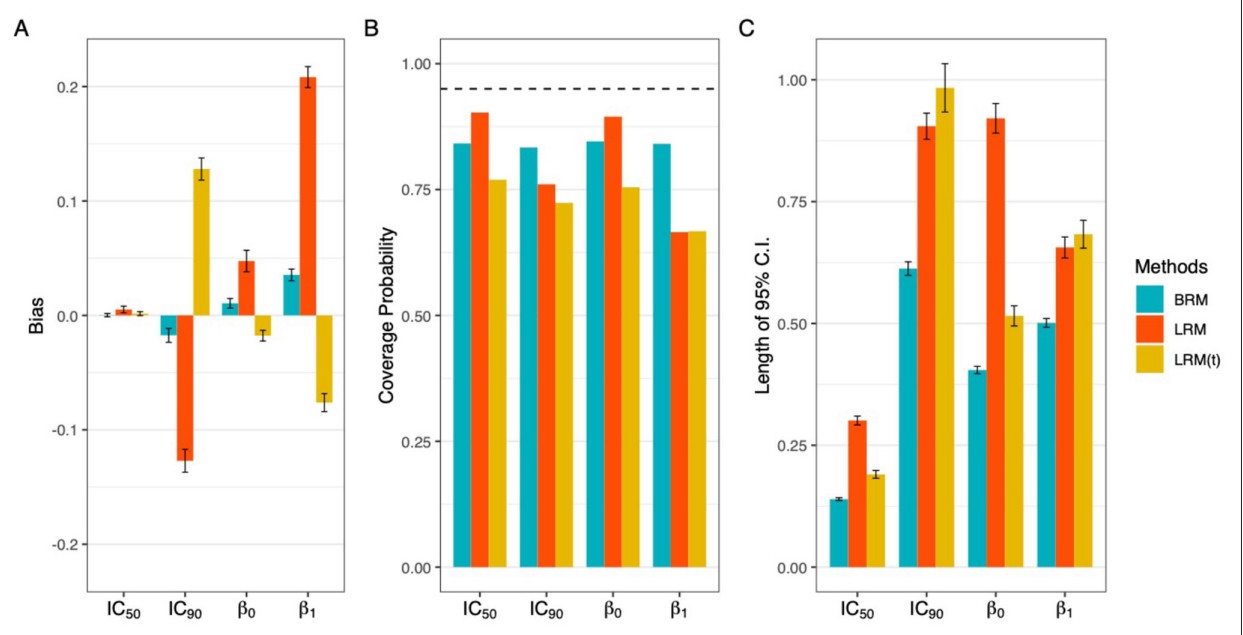

**Figure 2.** Comparison of estimation efficiency and accuracy using linear regression model and beta regression model. Deleting the extreme values could not eliminate the bias (panel A), but only harmed the accuracy with worse coverage probabilities (panel B) and impaired the efficiency of interval estimation with wider nominal 95% confidence intervals (panel C). A total of 1000 data sets were generated following the data simulating process described in **Appendix 1**, using the dose sets and true dose-response curve under 7 dose setting with a precision parameter of 100. Responses ≤5% or ≥95% were considered extreme responses. Dashed line in panel B denotes 95% nominal coverage probability. BRM: beta regression with extreme data points; LRM: linear regression model with extreme data points; LRM(t): linear regression model with truncated dataset after deleting extreme values. Detailed model descriptions of LRM and BRM are provided in Materials and methods section.

established in an agile modeling framework under the parameterization of the beta law to describe a continuous response variable with values in a standard unit interval (0.1). We further exploited a robust estimation method of the beta regression, named the minimum density power divergence estimators (MDPDE) (*Ghosh, 2019*), for dose-response estimation, with the tuning parameter optimized by a data-driven method (*Ribeiro and Ferrari, 2020*). The technical details are provided in the Materials and methods.

REAP presents a straightforward analytic environment for robust estimation of dose-response curve and assessment of key statistics, including implementation of statistical comparisons and delivery of customized output for graphic presentation (*Figure 3*). The dose-response curve is a time-honored tool to convey the pharmacological activity of a compound. Through dose-response curves, we can compare the relative activity of a compound on different assays or the sensitivity of different compounds on an assay. REAP aims to make this job simple, estimation efficient, and results robust.

There are three sections in REAP: Introduction, Dataset and Output. Users can have both overview and instruction of REAP in the Introduction. Dataset is uploaded in the Dataset section. The input dataset is mandated to be in a csv file format and contains three columns of data respectively for drug concentration, response effect and group name, in a specific order. It is recommended that users normalize the response variable to the range of (0,1) by themselves. Otherwise, REAP automatically will truncate the values exceeding the boundaries to (0,1) using a truncation algorithm (see Appendix 1 - Truncation Strategy). In the Output section, it generates a dose-response plot, along with tabulation for effect and model estimations. A special feature of REAP is that it conveniently allows the users to specify the target effect level, rather than fixed at the common median effect (i.e., 50%), in dose estimation. We also enable hypothesis testing for comparisons of effect estimations, slopes and models (i.e. comparing both intercepts and slopes; see Materials and methods). By default, the x-axis of the dose-response plot is log-scaled. In the plot, users can choose to add mean values and sample standard deviations for data points under the same agent and dose level. Both plots and estimation tables are downloadable on REAP to plug in presentations and manuscripts for result dissemination.

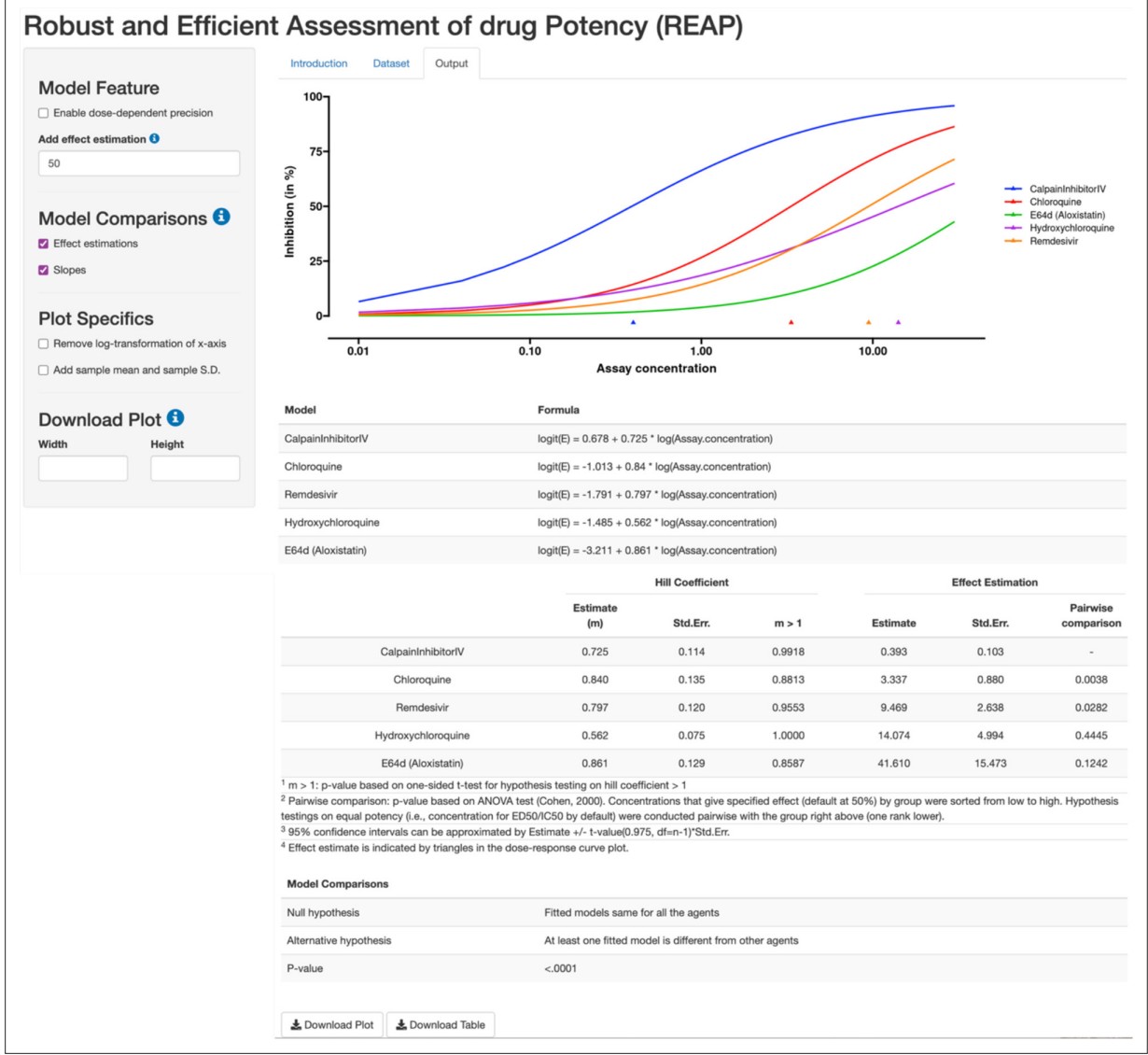

**Figure 3.** REAP App interface, with a highlight of Output section. Using the robust beta regression method, REAP produces a dose-response curve plot with effect and model estimations. The left panel allows users to specify model features and design plot specifics. REAP also provides hypothesis testing results to compare effect estimations, slopes and models.

The open-sourced REAP is freely available and accessible at https://xinying-fang.shinyapps.io/ REAP/. We demonstrated it in two real-world examples, after presenting the simulation results, to illustrate the functionality of REAP.

## Simulations

We conducted simulation studies to investigate the robust beta regression model, in comparison to linear regression models with data transformation, either under a normal distribution error (implemented with R package 'stats') or a heavy-tailed t distribution error with 3 degrees of freedom (implemented with R package 'heavy'), to characterize the median-effect equation under different scenarios. The model assessment is established based on both the point estimation and interval estimation derived from each method. Details on the simulation setting are described in the Appendix 1 - Data simulating process.

With data simulated using normal error terms, the robust beta regression provides sensible estimation of $IC_{50}$, $IC_{90}$, $\beta_1$, and $\beta_0$ from median-effect equation (*Figure 4*, *Appendix 1—table 1*). Particularly, when there are extreme outcome observations, the robust beta regression manages much

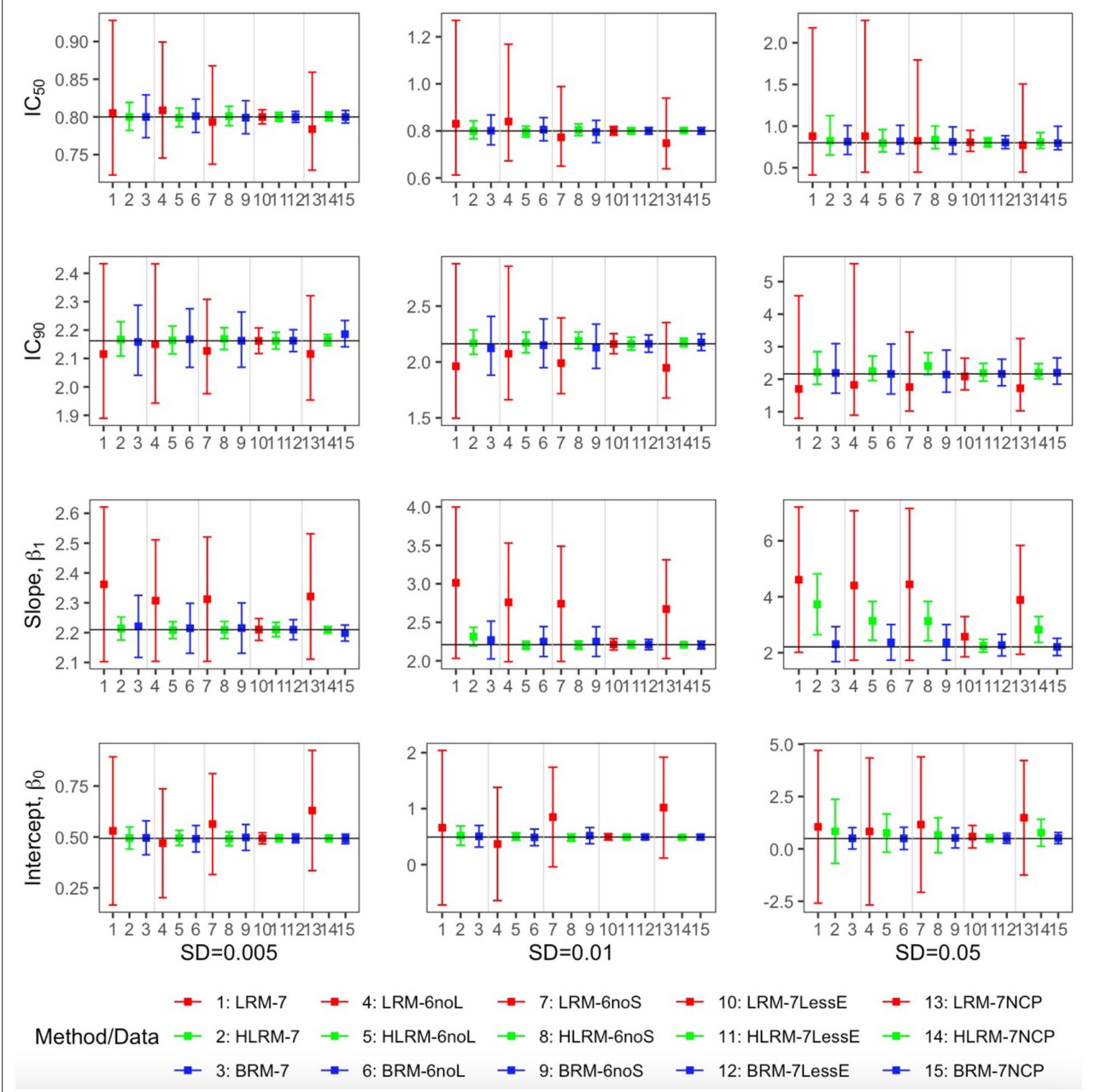

**Figure 4.** Comparison of the point estimates and 95% confidence intervals using linear regression model, heavy-tailed linear regression model and robust beta regression model, with data simulated from normal error term. The vertical solid lines indicate the true values. The dots represent the averaged point estimates and the bars represent the averaged lower and upper bound of 95% CIs. The point estimation by robust beta regression is consistently closer to the true value with a narrower 95% CI compared to the linear regression model. The 95% CI of heavy-tailed linear regression underestimates the nominal coverage probability. LRM: linear regression model; LRM-7: LRM under 7-dose dataset with extreme data points; LRM-6noL: LRM under 6 dose dataset after removing the highest dose data point; LRM-6noS: LRM under 6-dose dataset after removing the lowest dose data point; LRM-7lessE: LRM under 7-dose dataset with less extreme data points; LRM-7NCP: LRM under 7-dose dataset with extreme data points and dose-dependent precision; HLRM: heavy-tailed linear regression model; HLRM-7: Heavy-tailed LRM under 7-dose dataset with extreme data points; HLRM-6noL: Heavy-tailed LRM under 6-dose dataset after removing the highest dose data point; HLRM-6noS: Heavy-tailed LRM under 6-dose dataset after removing the lowest dose data point; HLRM-7lessE: Heavy-tailed LRM under 7-dose dataset with less extreme data points; HLRM-7NCP: Heavy-tailed LRM under 7-dose dataset with extreme data points and dose-dependent precision; BRM: robust beta regression model; BRM-7: BRM under 7-dose dataset with extreme data points; BRM-6noL: BRM under 6-dose dataset after removing the highest dose data point; BRM-6noS: BRM under 6-dose dataset after removing the lowest dose data point; BRM-7lessE: BRM under 7-dose dataset with less extreme data points; BRM-7NCP: BRM under 7-dose dataset with extreme data points and dose-dependent precision. Detailed model descriptions of LRM, HLRM, and BRM are provided in Materials and methods section.

lower bias and root-mean-square error (RMSE) for point estimates and better coverage probability for interval estimates than the linear regression model with normal distribution error. For data without extreme values, their performance is comparable in bias, RMSE and coverage probability, but the linear regression model has much wider 95% CIs (*Figure 4*). Indeed, the wider 95% CIs occur across all the scenarios, indicating higher estimation efficiency of the robust beta regression approach. In contrast, the heavy-tailed linear regression model demonstrates improved bias and RMSE in point estimation from the standard linear regression, but the nominal 95% CIs are significantly underestimated with coverage probability below 50% in most cases (*Appendix 1—table 1*). Therefore, the heavy-tailed linear regression model, although sometimes provides good point estimations, cannot maintain consistently robust and statistically efficient estimations. Overall, the robust beta regression model is the most robust and stable in estimating the median-effect equation with reliable performance in both point estimations and 95% CI coverage probabilities.

In parallel, similar results are obtained consistently with data simulated using beta error terms, which induces heteroscedasticity (smaller variation on the two ends and bigger in the middle) at different dose levels (*Appendix 1—figure 1*, *Appendix 1—table 2*). All the results above demonstrate the sensitivity of regression models in dealing with datasets including extreme values. In addition, the result comparisons between the seven-dose set and the six-dose set with the largest or smallest dose eliminated display the potential worse influence of deleting extreme values directly in modeling dose-response using linear regression, which further notarizes the robustness and efficiency of the proposed robust beta regression.

Overall, the simulation study suggests that the robust beta regression model produces well-calibrated dose-response curves while being more robust and powerful than the standard regression model and the heavy-tailed linear regression model in estimating the median effect equation.

## B-cell lymphoma data

The first example of REAP application is dose-response curve estimation of the same agent under different cell lines. The data was originally from a study on using a drug called auranofin in treating B-cell lymphomas such as relapsed or refractory mantle cell lymphoma (MCL) (*Wang et al., 2019*). As an FDA-approved treatment of rheumatoid arthritis, auranofin targets thioredoxin reductase-1 (Txnrd1), and was repurposed as a potential antitumor drug to effectively induce DNA damage, reactive oxygen species (ROS) production, cell growth inhibition, and apoptosis in aggressive B-cell lymphomas, especially in TP53-mutated or PTEN-deleted lymphomas.

In the experiment, the effect of auranofin was evaluated in six MCL cell lines (Z-138, JVM-2, Mino, Maver-1, Jeko-1, and Jeko-R) with auranofin in concentrations ranging from 0 to 5 μM for 72 hr and tested cell viability using a luminescent assay. The interval bars of observed dose-response in *Figure 5* show that the sample variance of error from repeated measurements decreased with the increase of auranofin concentrations. To account for the heteroscedasticity and asymmetry in the variance, we enable a dose-dependent precision (proportional to inverse variance) in REAP, adding $\log(dose)$ as an

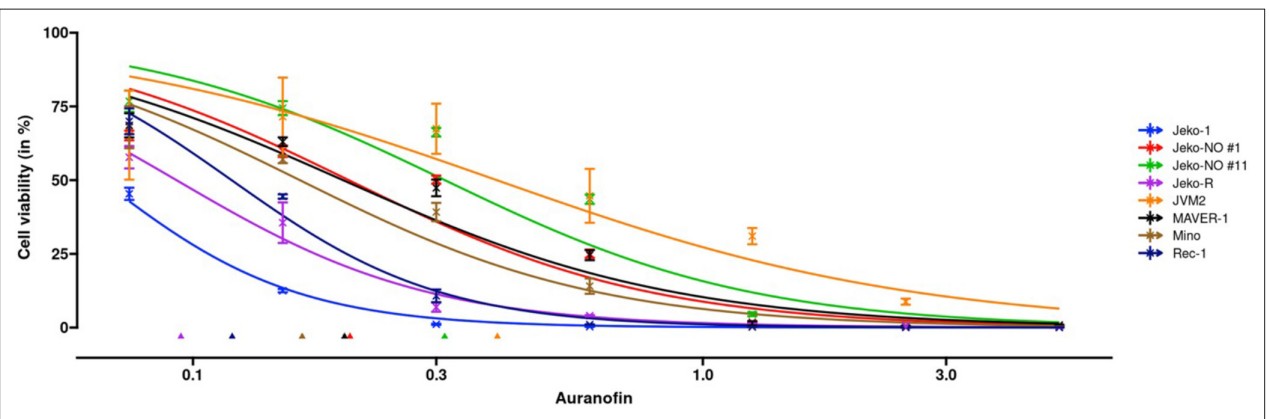

**Figure 5.** Dose-response curve estimation of auranofin (μM) under different MCL cell lines. The dose-response curve was fitted with a dose-dependent precision with $\log(dose)$ as an additional regressor for the precision estimator. Observed dose effects are displayed with interval bars, which end with arrows when estimated intervals exceed (0,1). Triangles at the bottom indicate $IC_{50}$ values for each MCL cell line. MCL: mantle cell lymphoma.

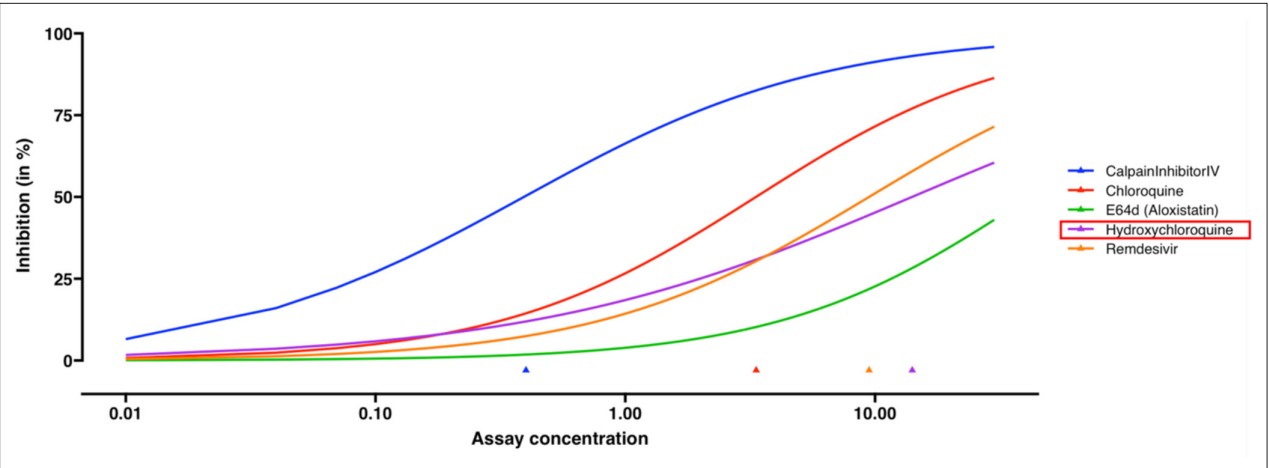

**Figure 6.** Dose-response curve estimation of anti-viral drugs under the same biological batch with SARS-CoV-2 data. The robust beta regression gives reasonable estimations to the dose-response curve of hydroxychloroquine, compared to the inconclusive dose-response curve fitted by linear regression in Bobrowski et al. (2020). The plot is generated without selecting the option of mean and confidence interval for observations. Triangles indicate the estimated $EC_{50}$ values for each drug.

additional regressor for the precision parameter. *Figure 5* shows the fitted dose-response curves with the dose-dependent precision. The test for homogeneity (p-value <0.0001) suggests distinct dose-response between cell lines. The estimation of intercepts, hill coefficients and pairwise comparisons of $IC_{50}$ estimations are provided in *Appendix 1—table 3*.

## SARS-CoV-2 data

The second example is on the dose-response curve estimation in antiviral drug development for coronavirus disease 2019 (COVID-19). At the beginning of 2020, COVID-19 broke out at an unprecedented pace internationally, but there were limited therapeutic options for treating this disease. Therefore, many compounds and their combinations were rapidly tested in vitro against the SARS-CoV-2 virus to identify potentially effective treatments and prioritize clinical investigation.

In the data (*Bobrowski et al., 2021*), the benchmark compound collection consists of five known antivirals, including remdesivir, E64d (aloxistatin), chloroquine, calpain Inhibitor IV and hydroxychloroquine. The in vitro experiment was performed using the same biological batch of SARS-CoV-2 virus and conducted in biosafety level-3. In the original publication (*Bobrowski et al., 2021*), the dose-response curves were fitted by linear regression, which could yield inconclusive estimation (e.g. hydroxychloroquine in Figure 1G of *Bobrowski et al., 2021*), while the estimated inhibition tends to exceed 1 when concentration is larger than 10 µM. REAP gives reasonable estimation for the dose-response curves (*Figure 6*). The hypothesis testing results show that at least one slope estimation is different from other antivirals (p-value = 0.0003) and at least one $EC_{50}$ estimation is different from others (p-value = 0.003). Calpain Inhibitor IV shows a higher potency than other agents including hydroxychloroquine (p-value = 0.0038, *Appendix 1—table 4*).

## Discussion

Quantifying the potency of a compelling substance is always a central topic in life sciences (*Schindler, 2017*). It is a vital component of research in pharmacology, but also prevalent in the fields of toxicology, environmental science, agrochemistry, and medicine, among many others. For instance, the description of dose-response curves can provide the initial toxicological risk assessment (*National Research Council, 2007*), and guide in silico modeling of toxic doses to humans and the environment (*Blaauboer et al., 2012*). Based on proper identification of dose-response relationship from in vitro assays, studies can successfully predict systemic toxicological effects in vivo without additional in silico modelling (*Groothuis et al., 2015*). Nevertheless, it necessitates accurate and reliable description of the dose-response curve, which further demands robust and efficient modeling strategies to account

for embedded variability in observed response and to derive solid inference with valid quantification of uncertainty.

The dose-response estimation could be substantially biased by the standard regression modeling. In the illustrative example (*Figure 1*), the estimated $IC_{50}$ dose indeed effects the 70% fraction of cell affected, while the estimated response at the true $IC_{50}$ dose is only 22%. Such a large discrepancy is sourced by a small (<0.5%) single measurement error, which is common and inevitable in any regular in vivo experiment, but could engender a profound impact on the assessment of drug potency and determination of synergy in drug combinations. In addition, the modeling strategy of deleting those extreme values (e.g. *Figure 2*, or 6noL and 6 noS datasets in *Figure 4* and *Appendix 1—figure 1*) is futile to improve the poor performance of standard regression model, but may further impair the estimation efficiency and accuracy. In general, it fails to reduce bias but only introduces larger uncertainty in estimation of dose concentration, especially at extreme responses (e.g. $IC_{90}$). On the other hand, a heavy-tailed error distribution may help to stabilize the point estimation, but the interval estimation could be largely under-estimated with poor coverage probabilities.

We develop REAP for assessment of drug potency to address concerns in this regard. It has substantial advantages over existing methods by reducing the impact of random errors due to implicit variations in the experimental data. To our best knowledge, it is also for the first time that beta regression is introduced to dose-response estimation. The underlying modified robust beta regression model estimated by the data-driven tuning parameter is resilient to estimation bias caused by extreme observations, which is a routinely encountered situation for deficient dose-response estimation using the standard estimation approach. The proposed approach is also efficient in quantitative characterization of dose-response curves with narrower confidence intervals for key estimators. Furthermore, REAP can simultaneously model the data heterogeneity with a dose-dependent precision component (*Figure 5*). It is simply different from other dose-response methods, in which a vector of weights have to be (possibly mis-)specified externally. REAP is an open-source and user-friendly platform, developed for diverse non-computational scientists for hands-on wet-laboratory data analysis in regular use, and can be hosted within R shiny environment under Windows, Linux, and Mac systems or deployed in Docker available as a web server.

Our work potentially can be useful in applications of drug screening. The proposed method and the developed REAP App allow for the robust and efficient estimation and accounting for outliers as well, making it fitted particularly in a high-throughput setting. As the result of a complex and dynamic cascade of events, exposure time is another important factor ultimately affecting the dose-response. For in vitro experiments measured at different time points in a choice of cell-lines and expressed by a variety of assays (*Byrne and Maher, 2019*), the proposed modeling framework can be naturally extended to model time-dependent cytotoxicity while controlling for fixed or random effects. Furthermore, the application of robust and efficient dose-response estimation can be integrated into methods to identify drug interaction effect (*Lee and Kong, 2009*; *Lee et al., 2007*). There is a venerable history that multi-agent combination therapies demonstrate great advantages in improving therapeutic efficacy and revolutionize patient outcomes in a wide range of diseases. Robust and efficient estimation of the dose-response curve would be crucial in investigation of adequate drug combinations.

The developed method has limitations. We presented a model of the median effect equation for dose-response curve estimation based on mass action law. While in specific scenarios other laws may be considered more suitable to describe the biomedical systems, the current modeling framework can be naturally adapted for other dose-response functions like probit (via cumulative normal distribution) and Weibull model (*Christensen, 1984*), or any other continuous distribution functions. In addition, the median-effect equation to characterize pharmacological activity assumes the compound can affect all the cells. From a quantitative perspective, a compound that cannot reach high binding affinity will yield an over-conservative estimation for median effective dose of a drug. However, in comparison to the sensitivity of different compounds in an assay, it is not harmful because the less effective compounds will be more easily identified. If it is a concern that the maximal effects of candidate compounds are different and the aim is to accurately model the dose-response curve, the Emax model could be a better choice (*Lee et al., 2010*). Furthermore, the robust beta regression approach in REAP cannot handle values equal or less than 0, or equal or greater than 1. Thus, we developed a sequential data truncation algorithm in REAP to overcome the limitation of the conventional transformation ($y * (n−1)+0.5) / n$, which could be too rough in dose-response curve estimation particularly

when the sample size $n$ for each group is relatively small. Although empirically we have validated it using simulated data, the algorithm could be improved by future work to retain information more efficiently.

In summary, a good modeling strategy must effectively characterize the nature of the observed dose-response pattern (*Lyles et al., 2008*). Rapid advances in novel drug development and considerable deficiency in modeling data with extreme values offer an appealing opportunity for next-generation quantitative approaches. While many aspects of the techniques discussed here fit in the statistical framework of robust beta regression, our aim is to clearly apply and rigorously customize the analytic considerations, to reduce bias and ameliorate efficiency in routinely used dose-effect estimation, and to facilitate the convenient analytic implementation and dissemination. Experimental conditions and candidate drug potency could inevitably vary in practice, but REAP provides a great tolerance for points with extreme values, solid support for accurate and efficient dose-response curve estimation, and useful reference to the future development of methodology in drug investigation. Overall, we anticipate that our work will contribute more to quantitative analysis in assessment of drug potency in preclinical research.

## Materials and methods
### Median-effect equation and dose-response curve

The median-effect equation describes a popular model of the dose-response relationship based on the median effect principle of the mass action law in various biological systems (*Chou, 1976*). Assume $f_a$ and $f_u$ are the fractions of the system affected and unaffected by a drug concentration $d$. The median-effect equation states that

$$\frac{f_a}{f_u} = \left(\frac{d}{D_m}\right)^m,\tag{1}$$

where $m$ is the Hill coefficient signifying the sigmoidicity of the dose-effect curve and $D_m$ is the dose of a drug required to produce the median effect, which is analogous to the more familiar $IC_{50}$ (drug concentration that causes 50% of the maximum inhibitory effect), $ED_{50}$ (half-maximum effective dose), or $LD_{50}$ (median lethal dose) values (*Ghosh, 2019*). For example, if an inhibitory substance is of interest, the parameter $m$ measures the cooperativity in the binding of multiple ligands to linked binding sites, and the parameter $D_m = IC_{50}$, defined by the concentration that causes 50% of the maximum inhibitory effect.

Given $f_a + f_u = 1$, the median-effect *Equation 1* is equivalent to

$$\text{logit}\left(f_a\right) = \log\frac{f_a}{f_u} = -\text{logit}\left(f_u\right) = -\log\frac{f_u}{f_a} = m\left(\log d - \log D_m\right),\tag{2}$$

where $\text{logit}\left(p\right)$ denotes the logit function $\log\frac{p}{1-p}$. The *Equation 2* shows a log-linear relationship between the drug dose $d$ and its effect $f_a$ (or $f_u$, if it is, for example, the % survival of interest) after a logit transformation. Because from a modeling perspective the identical strategy can be applied to model both $f_a$ and $f_u$, for the effect on cell fraction $E$, we can rewrite *Equation 2* to be:

$$\text{logit}\left(E\right) = \log\frac{E}{1-E} = \beta_1 \log d + \beta_0\tag{3}$$

where $\beta_0$ is the intercept and $\beta_1$ the slope of the response curve. A linear regression model (LRM) can be applied in the form of *Equation 3* with a standard normal distribution error. In simulation studies, we also examine *Equation 3* with a heavy-tailed t-distribution error, denoted by heavy-tailed linear regression model (HLRM).

In this presentation, the median effect dose

$$D_m = exp\left(-\frac{\beta_0}{\beta_1}\right),\tag{4}$$

the Hill coefficient

$$m = \begin{cases} \beta_1 & \quad E = f_a \\ -\beta_1 & \text{if} \quad E = f_u \end{cases}\tag{5}$$

and the dose-response curve

$$E = \text{logit}^{-1} \left( \beta_1 \log d + \beta_0 \right), \tag{6}$$

where $\text{logit}^{-1}(x) = \frac{\exp(x)}{1+\exp(x)}$ is the inverse-logit function.

## Beta regression model for dose-response curve estimation

We will review the beta regression model which for the first time will be applied in dose-response estimation. The effect $E$ and the parameters $\beta = (\beta_0, \beta_1)$ in *Equation 3* cannot be directly observed, but they can be estimated using experimental data, in which the observed sample cell fraction $y$ produced by the drug dose $d$ is a random variable with mean $E$. It is clear that effective estimation must properly account for random variation and be based upon a model that not only matches the nature of the response variable, but adequately characterizes the observed dose-response pattern (*Lyles et al., 2008*).

Among all the unknown quantities, the parameters $\beta$ could be first estimated and play a fundamental role in supporting the inference for others. In the standard estimation procedure based on linear regression, $\text{logit}(y) = \log \frac{y}{1-y}$ is regressed on $\log d$ to get the inference on parameters $\beta$. Subsequently, the dose-response curve can be estimated by *Equation 6*, and $(D_m, m)$ can be derived based on *Equations (4) and (5)* for median-effect *Equation 2*. Because the extreme values of $y$ close to 0 or 1 could yield very large values of $\text{logit}(y)$ (approaching to $-\infty$ or $+\infty$, respectively, if $y \to 0$ or 1), and induce significant bias in estimation of $\beta$, the accuracy of the estimated dose-response curve and median-effect equation is in question when there exist extreme values in the dataset.

The beta regression model describes a response variable $y$ with continuous values restricted to the open standard unit interval (*Johnson et al., 1995*; *Simas et al., 2010*). In a classic beta regression framework, the beta regression model uses a parameterization of the beta law that is indexed by the mean parameter μ, and the precision parameter $\phi$ that controls the overall variation (*Ferrari and Cribari-Neto, 2004*). To model the dose-response relationship for the cell fraction $E$, we assume that the response $y$ is a beta-distributed random variable and its mean $\mu = E$ has the form of *Equation 6*, where $d$ is the dose producing effect $E$, $\beta_1$ and $\beta_0$ are the regression parameters. Estimation of regression parameters $\beta$ can be performed using maximum likelihood method to derive point estimate $\beta$ and covariance matrix $\Sigma$.

Beta regression is resistant to extreme values and provides reliable estimations (*Figure 1*). Compared with the standard approach, which applies a non-linear transformation in the response for an approximation to the normal distribution, the beta density can take on a variety of shapes to account for non-normality and skewness (*Smithson and Verkuilen, 2006*). In the presence of heteroskedasticity and asymmetry, two common problems frequently observed in limited range continuous response data, an empirical study showed that the beta regression provided the best estimation among several alternatives (*Kieschnick and McCullough, 2016*).

## Robust beta regression model with MDPDE

We will present a modified robust beta regression approach in REAP implementation, which is established based on density power divergence for robust estimation (*Ghosh, 2019*), but further improved after we introduce a data-driven method to identify the optimal tuning parameter. The standard beta regression potentially could still be sensitive against outliers because its inference is based on the maximum likelihood estimation. *Ghosh, 2019* developed the robust minimum density power divergence estimators (MDPDE) that address the problem by minimizing the average density power divergence (DPD)

$$d_\alpha(\hat{g}, g) = \int g^{1+\alpha} - \frac{1+\alpha}{\alpha} \int \hat{g} g^\alpha + \frac{1}{\alpha} \int \hat{g}^{1+\alpha},$$
$$d_0(\hat{g}, g) = \lim_{\alpha \to 0} d_\alpha(\hat{g}, g) \int \hat{g} \log \left( \frac{\hat{g}}{g} \right), \tag{7}$$

between the empirical density $g$ and the beta model density function $g \equiv Beta\left(\mu\phi, (1-\mu)\phi\right)$ with $\mu = \text{logit}^{-1}\left(\beta_1 \log d + \beta_0\right)$. $\alpha$ is a non-negative tuning parameter, smoothly connecting the likelihood disparity (at $\alpha = 0$) to the $L_2$-Divergence (at $\alpha = 1$). The parameter of interest $\beta$ is estimated by minimizing the DPD measure between $g_i$ and the density, $g_i$,

$$n^{-1} \sum_{i=1}^{n} d_\alpha(\hat{g}_i(\cdot),\ g_i(\cdot, \boldsymbol{\theta})) \tag{8}$$

where $\boldsymbol{\theta} = (\beta, \phi)^T$. After mathematically simplifying *Equation 8*, (*Ghosh, 2019*), $\boldsymbol{\theta}$ can be equivalently estimated by minimizing the objective function using the estimation equations:

$$H_{n,\alpha}\left(\boldsymbol{\theta}\right) = n^{-1} \sum_{i=1}^{n} \left[ K_{i,\alpha}\left(\boldsymbol{\theta}\right) - \tfrac{1+\alpha}{\alpha} g_i\left(y_i, \boldsymbol{\theta}\right)^\alpha \right] \tag{9}$$

where $K_{i,\alpha}\left(\boldsymbol{\theta}\right) = \frac{B\left((1+\alpha)\mu_i\phi,\ (1+\alpha)\left(1-\mu_i\right)\phi - \alpha\right)}{B\left(\mu_i\phi,\ (1-\mu_i)\phi\right)^\alpha}$.

MDPDE improves the standard beta regression with the DPD measure and a fixed tuning parameter. The recommended $\alpha$ is around 0.3 to 0.4, but simply assigning a fixed $\alpha$ in [0.3, 0.4] is not applicable in many cases. Here we adopted a data-driven method (*Ribeiro and Ferrari, 2020*) to identify the optimal $\alpha$. The search for the optimal $\alpha$ starts with a grid of $\alpha$, a pre-defined $\alpha_{max}$ and grid size $\rho$, which generates a sequence of equally spaced $\{\alpha_k\}_{k=0}^{m}$ $(0 = \alpha_0 < \alpha_1 < \cdots \alpha_m \leq \alpha_{max})$. MDPDE calculates the corresponding $\boldsymbol{\theta}$ and se($\boldsymbol{\theta}$) with each $\alpha$ so that we get a vector of standardized estimates:

$$z_{\alpha_k} = \left( \frac{\widehat{\theta}_{\alpha_k}^1}{\sqrt{n}se\left(\widehat{\theta}_{\alpha_k}^1\right)},\ \ldots,\ \frac{\widehat{\theta}_{\alpha_k}^p}{\sqrt{n}se\left(\widehat{\theta}_{\alpha_k}^p\right)} \right)^T$$

The standardized quadratic variations (SQV) are defined by:

$$SQV_{\alpha_k} = p^{-1} \| z_{\alpha_k} - z_{\alpha_{k+1}} \|.$$

We compare each $SQV_{\alpha_k}$ with a pre-defined threshold $L$ $(L > 0)$. If all $\alpha_k$ satisfy the stability condition of $SQV_{\alpha_k} < L$, then the optimal $\alpha$ equals the minimal $\alpha$ in $\alpha_k$ . Otherwise, restart the search with a new grid of $\alpha_k$ . The new grid of the same size $p$ is picked from the sequence $\{\alpha_k\}_{k=0}^{m}$ starting from the largest $\alpha_k$ that fails the stability condition. Repeat searching until all $\alpha_k$ in the current grid satisfy the stability condition or $\alpha_{max}$ is reached. If the stability condition is satisfied before $\alpha_{max}$ is reached then optimal $\alpha$ equals the minimal value in the grid of $\alpha_k$ . If $\alpha_{max}$ is reached, then optimal $\alpha$ equals 0, which is equivalent to the maximum likelihood estimation. We denote this approach by robust beta regression model (BRM) in the simulation study.

## Point estimate and its confidence interval for drug activity measurements

The objective of analysis is to characterize the dose-response curves in *equation (2)* and quantify in vitro drug potency. Popular drug activity measurements include Hill coefficient $m$ and median effect dose $D_m$ . In some circumstances, other measurements such as instantaneous inhibitory potential (*IIP*), which directly quantifies the log decrease in single-round infection events caused by a drug at a clinically relevant concentration, are of special interest (*Shen et al., 2009*).

The MDPDE for beta regression model provides a robust strategy to estimate $\beta$, from which the point estimates and confidence intervals of relevant drug activity measurements can be derived. Mathematically, those drug activity quantities can be written as functions of parameters $\beta$ with an explicit form. Subsequently, their point estimates and confidence intervals can be derived based on the inference of $\beta$. For example, given a point estimate $\hat{\beta} = (\hat{\beta}_0, \hat{\beta}_1)$, the point estimate for $\hat{m}$ , $\hat{D}_m$ as a single value, and $\hat{E}$ as a function of dose $d$ can be computed using *Equations 4–6*.

It is important to construct the confidence interval around the point estimate to gauge the estimation uncertainty. With different levels of measurement error from either well-managed or lousy experiments, the levels of evidence vary for statistical inference, even if it derives the same point estimates for the intercept $\beta_0$ , slope $\beta_1$ and the corresponding dose-response curve. Given the point estimate $\hat{\beta}$ and its positive-definite covariance matrix $\Sigma$ to account for variability in observed response, we apply the multivariate delta method and approximate the variance estimate after assuming asymptotic normality (*Bickel and Doksum, 2015*). As demonstrated in our simulation studies, the constructed $(1 - \alpha) \times 100\%$ confidence interval consistently provides better results to quantify the $(1 - \alpha) \times 100\%$ coverage probability. More importantly, the width of the constructed confidence interval was narrower

than that from a linear regression model, suggesting that our approach is more efficient with a higher statistical power (*Appendix 1—tables 1 and 2*).

## Comparison of the dose-response curves

When we estimate multiple dose-response curves with the data collection experiments conducted in a similar setting, it is often of interest to statistically compare the drug potency and/or Hill coefficients. A typical comparison may occur when we examine the similarity of response from different drugs, explore the additional effect of a drug combined with certain monotherapy, or assess the homogeneity of a drug to different patient samples or cell lines. In the beta regression framework, the statistical comparison can be conducted by first comparing independent fits for each curve with a global fit that shares the common parameters among different groups. Subsequently, the likelihood ratio test can be applied to examine whether the same Hill coefficient or one dose-response curve can adequately fit all the data. The only exception is to assess whether median effect doses are the same in different groups, while an F test is used for the single parameter testing. If the global test for potency shows a significant p-value, a pairwise comparison can be conducted using two-sided t-test for the ordered groups with Benjamini-Hochberg correction for multiplicity.

## Acknowledgements

This study was supported in part by the Penn State College of Medicine Junior Faculty Development Award, MD Anderson B-cell Lymphoma Moon Shot Program, and NIH Cancer Center Support Grant P30 CA016672.

## Additional information

### Funding

| Funder | Grant reference number | Author |
|--------|------------------------|--------|
| Penn State College of Medicine | Junior Faculty Development Award | Shouhao Zhou Xinyi Liu |
| University of Texas MD Anderson Cancer Center | B-cell Lymphoma Moonshot Program | Shouhao Zhou |
| National Cancer Institute | Cancer Center Support Grant P30 CA016672 | Shouhao Zhou J Jack Lee |

The funders had no role in study design, data collection and interpretation, or the decision to submit the work for publication.

### Author contributions

Shouhao Zhou, Conceptualization, Funding acquisition, Investigation, Methodology, Project administration, Software, Supervision, Writing – original draft, Writing – review and editing; Xinyi Liu, Data curation, Formal analysis, Investigation, Methodology, Visualization, Writing – original draft; Xinying Fang, Data curation, Formal analysis, Investigation, Methodology, Software, Visualization, Writing – original draft; Vernon M Chinchilli, Investigation, Resources, Writing – review and editing; Michael Wang, Nikolay V Dokholyan, Funding acquisition, Investigation, Writing – review and editing; Hong-Gang Wang, Chan Shen, Investigation, Writing – review and editing; J Jack Lee, Conceptualization, Funding acquisition, Investigation, Writing – review and editing

### Author ORCIDs

Shouhao Zhou http://orcid.org/0000-0002-8124-5047
Xinying Fang http://orcid.org/0000-0001-9121-5717
Nikolay V Dokholyan http://orcid.org/0000-0002-8225-4025

### Decision letter and Author response

Decision letter https://doi.org/10.7554/eLife.78634.sa1
Author response https://doi.org/10.7554/eLife.78634.sa2

## Additional files

### Supplementary files
• MDAR checklist

### Data availability
All datasets in illustrative examples are available at https://github.com/vivid225/REAP/blob/main/REAP/ to generate the figures and tables using REAP.

The following dataset was generated:

| Author(s) | Year | Dataset title | Dataset URL | Database and Identifier |
|---|---|---|---|---|
| vivid225 | 2022 | REAP | https://github.com/vivid225/REAP/tree/main/REAP | GitHub, 15973b5 |

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

# Appendix 1

## Truncation strategy

Based on the median-effect equation method by Chou TC, the software "CompuSyn" was published. In the data entry illustration of this software, they pointed out the sensitivity limits of data points, for example too low (fa <0.02) and too high (fa >0.99) and suggested that such data points out of effect may be edited or deleted.

There are some data truncation algorithms in the literature. Two obvious remedies are proportionally "shrinking" the range to a sub-range nearly covering the unit interval (e.g., [.00001,.99999]) or simply adding a small amount to 0-valued observations and subtracting the same amount from 1-valued observations while leaving the other observations unchanged. *Macmillan and Creelman, 2005* mentioned a method that is frequently used in practice in areas such as signal detection is to add 1/(2n) to a 0 observation and subtract 1/(2n) from a 1 observation, where n is the total number of observations. Besides, *Smithson and Verkuilen, 2006* demonstrated that a useful transformation in practice is (y * (n−1)+0.5) / n, which is also mentioned by the documentation for R Betareg package for conditions when data assumes the extremes 0 s and 1 s. In dose-response curve estimation, this treatment could be too rough, especially when n is small.

To minimize the impact from truncation of data points, we apply the following algorithm. The first step is to shrink the data range to [1e-9, 1-1e-9]. If there still exist abnormal conditions, we will sequentially shrink the data range of abnormal ones to [1e-8, 1-1e-8], …, until [1e-3, 1-1e-3] or non-exist of abnormal conditions. Then, if it still exists, though rarely, the transformation of (y * (n−1)+0.5) / n, where n is the sample size, in the documentation of R Betareg package would be applied. We have conducted simulations to test this algorithm in various scenarios with different errors and it achieved reasonable performance in handling all conditions.

## Data simulating process

In the simulation study, both robust beta regression and linear regression are applied to estimate dose-response curves under different scenarios. The point estimations and 95% confidence intervals of $IC_{50}$, $IC_{90}$, $\beta_0$ and $\beta_1$ under each method will be obtained and then, be compared to evaluate the model performance.

To generate data for simulation studies, we define the dose set for simulation as 0.1, 0.2, 0.4, 0.8, 1.6, 3.2 and 6.4 $\mu M$, which consists of 7 doses, and choose the appropriate true curve with $\beta_1 = 2.2098$ and $\beta_0 = 0.4931$ such that the corresponding effects of the smallest and largest dose are 0.01 and 0.99, respectively. Let's call the true curve "$E = f\left(\log\left(dose\right)\right)$". Then, the following equation is applied to generate data by inducing random error into effect:

$$E = true + error = f\left(\log\left(dose\right)\right) + error$$

We simulated data with two types of errors, normal error term and beta error term, to examine the accuracy and sensitivity of model performance in general setting. The normal error term is implemented with different standard deviations (SDs), for example 0.005, 0.01 and 0.05, while the beta error term is under different precision parameter $\phi$, for example 35, 15, 5. Note that the larger the $\phi$, the smaller the variance. By implementing under different SD or $\phi$, it allows for generation of not only well-controlled data which is assumed for experiments with almost no error, but also noised data which is more identical to real-world data. The generated data is 1 replicate given each dose level with the total simulation size equal 10,000 for each choice of SD or $\phi$. Since the defined dose set is symmetric, we set up several scenarios under both error terms above: (1) full 7-dose set with extreme values; (2) 6-dose set after removing the largest dose; (3) 6-dose set after removing the smallest dose; (4) full 7-dose set with less extreme values by obtaining the smallest and largest dose levels with corresponding effect as 0.1 and 0.9 under the same true curve. The scenarios 1–4 assume constant precision parameter during data simulation and modeling process.

To mimic the real-world environment of data collections, the assumption of equal variance doesn't always hold. Thus, we also set up the 5th scenario which uses full 7-dose set with extreme values with non-constant SD or precision parameter during data simulation and modeling process, but linearly dose-dependent. For normal error term, the modified SDs for data simulation have the form of $SD^* = \left(\gamma_0 + \gamma_1 * log.dose\right) * SD$; for beta error term, the modified precisions $\phi$ for data simulation

have the form of $\phi^* = (\gamma_0 + \gamma_1 * log.dose) * \phi$. Assuming the same true dose-response curve as the previous simulation, we pre-defined $\gamma_0$ and $\gamma_1$ as 0.25 and 0.1378 such that the average of $SD^*$ is close to $SD$, and the average of $\phi^*$ is close to $\phi$, respectively.

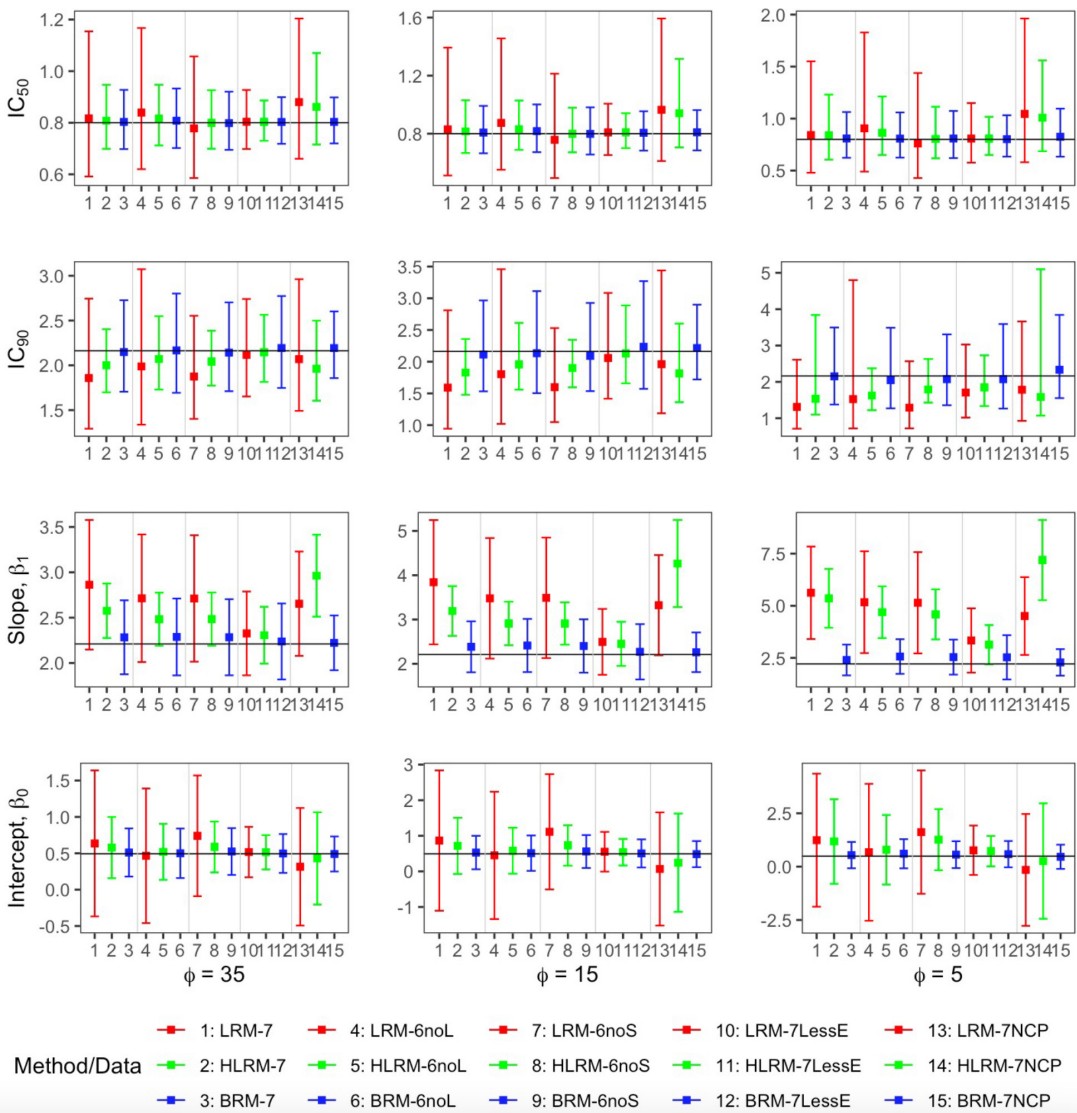

**Appendix 1—figure 1.** Comparison of the point estimates and 95% confidence intervals using linear regression model, heavy-tailed linear regression model and robust beta regression model, with data simulated from beta error term. The vertical solid lines indicate the true values. The dots represent the averaged point estimates and the bars represent the averaged lower and upper bound of 95% CIs. The point estimation by robust beta regression is consistently closer to the true value with a narrower 95% CI compared to the linear regression model. The 95% CI of heavy-tailed linear regression underestimates the nominal coverage probability. LRM: linear regression model; LRM-7: LRM under 7 dose dataset with extreme data points; LRM-6noL: LRM under 6 dose dataset after removing the highest dose data point; LRM-6noS: LRM under 6 dose dataset after removing the lowest dose data point; LRM-7lessE: LRM under 7 dose dataset with less extreme data points; LRM-7NCP: LRM under 7 dose dataset with extreme data points and dose-dependent precision; HLRM: heavy-tailed linear regression model; HLRM-7: Heavy-tailed LRM under 7 dose dataset with extreme data points; HLRM-6noL: Heavy-tailed LRM under 6 dose dataset after removing the highest dose data point; HLRM-6noS: Heavy-tailed LRM under 6 dose dataset after removing the lowest dose data point; HLRM-7lessE: Heavy-tailed LRM under 7 dose dataset with less extreme data points; HLRM-7NCP: Heavy-tailed LRM under 7 dose dataset with extreme data points and dose-dependent precision; BRM: robust beta regression model; BRM-7: BRM under 7 dose dataset with extreme data points; BRM-6noL: BRM under 6 dose dataset after removing the highest dose data point; BRM-6noS: BRM

*Appendix 1—figure 1 continued on next page*

*Appendix 1—figure 1 continued*
under 6 dose dataset after removing the lowest dose data point; BRM-7lessE: BRM under 7 dose dataset with less extreme data points; BRM-7NCP: BRM under 7 dose dataset with extreme data points and dose-dependent precision.

**Appendix 1—table 1.** Simulation result of bias, RMSE and 95% CI coverage probability corresponding to normal error terms.

| Scenario | Method | Bias | | | | RMSE | | | | 95% CI Coverage Probability | | | |
|---|---|---|---|---|---|---|---|---|---|---|---|---|---|
| | | $IC_{50}$ | $IC_{90}$ | $\beta_0$ | $\beta_1$ | $IC_{50}$ | $IC_{90}$ | $\beta_0$ | $\beta_1$ | $IC_{50}$ | $IC_{90}$ | $\beta_0$ | $\beta_1$ |
| (a)data simulated using normal error term with SD = 0.005 | | | | | | | | | | | | | |
| 7 doses with extreme values | LRM | 0.005 | −0.047 | 0.037 | 0.152 | 0.098 | 0.298 | 0.525 | 0.557 | 0.954 | 0.773 | 0.943 | 0.666 |
| | HLRM | 0.000 | 0.004 | 0.002 | 0.004 | 0.018 | 0.102 | 0.063 | 0.169 | 0.468 | 0.335 | 0.463 | 0.225 |
| | BRM | 0.000 | −0.004 | 0.003 | 0.011 | 0.013 | 0.130 | 0.044 | 0.088 | 0.981 | 0.927 | 0.975 | 0.889 |
| 6 doses after removing largest | LRM | 0.009 | −0.01 | −0.024 | 0.098 | 0.045 | 0.184 | 0.129 | 0.517 | 0.967 | 0.921 | 0.971 | 0.691 |
| | HLRM | −0.001 | 0.001 | 0.002 | −0.001 | 0.014 | 0.063 | 0.037 | 0.065 | 0.453 | 0.408 | 0.467 | 0.250 |
| | BRM | 0.001 | 0.005 | −0.00 | 0.005 | 0.009 | 0.132 | 0.024 | 0.089 | 0.955 | 0.892 | 0.950 | 0.844 |
| 6 doses after removing smallest | LRM | −0.007 | −0.036 | 0.070 | 0.102 | 0.037 | 0.306 | 0.362 | 0.533 | 0.969 | 0.623 | 0.933 | 0.695 |
| | HLRM | 0.001 | 0.007 | −0.002 | −0.001 | 0.014 | 0.083 | 0.046 | 0.064 | 0.456 | 0.249 | 0.400 | 0.259 |
| | BRM | −0.001 | 0.000 | 0.005 | 0.005 | 0.009 | 0.151 | 0.039 | 0.089 | 0.956 | 0.872 | 0.940 | 0.853 |
| 7 doses with less extreme values | LRM | 0.000 | −0.000 | 0.000 | 0.001 | 0.005 | 0.030 | 0.016 | 0.026 | 0.891 | 0.828 | 0.883 | 0.799 |
| | HLRM | 0.000 | 0.000 | 0.000 | 0.000 | 0.005 | 0.031 | 0.016 | 0.028 | 0.668 | 0.599 | 0.661 | 0.567 |
| | BRM | 0.000 | 0.000 | 0.000 | 0.000 | 0.004 | 0.026 | 0.013 | 0.024 | 0.864 | 0.823 | 0.860 | 0.800 |
| 7 doses with extreme values and dose-dependent precision | LRM | −0.016 | −0.047 | 0.137 | 0.111 | 0.069 | 0.409 | 0.583 | 0.520 | 0.965 | 0.573 | 0.921 | 0.685 |
| | HLRM | 0.000 | 0.003 | −0.001 | −0.001 | 0.008 | 0.049 | 0.027 | 0.036 | 0.325 | 0.188 | 0.290 | 0.151 |
| | BRM | 0.000 | 0.023 | −0.003 | −0.011 | 0.003 | 0.196 | 0.023 | 0.095 | 0.957 | 0.849 | 0.931 | 0.794 |
| (b)data simulated using normal error term with SD = 0.01 | | | | | | | | | | | | | |
| 7 doses with extreme values | LRM | 0.030 | −0.200 | 0.166 | 0.804 | 0.224 | 0.544 | 1.234 | 1.305 | 0.958 | 0.699 | 0.950 | 0.716 |
| | HLRM | 0.000 | 0.006 | 0.025 | 0.105 | 0.031 | 0.238 | 0.199 | 0.795 | 0.408 | 0.286 | 0.404 | 0.194 |
| | BRM | 0.001 | −0.039 | 0.013 | 0.060 | 0.030 | 0.172 | 0.094 | 0.155 | 0.981 | 0.931 | 0.976 | 0.892 |
| 6 doses after removing largest | LRM | 0.040 | −0.088 | −0.125 | 0.549 | 0.101 | 0.280 | 0.314 | 1.245 | 0.966 | 0.924 | 0.972 | 0.734 |
| | HLRM | −0.004 | 0.007 | 0.008 | −0.008 | 0.027 | 0.125 | 0.075 | 0.123 | 0.424 | 0.384 | 0.445 | 0.241 |
| | BRM | 0.005 | −0.012 | −0.007 | 0.040 | 0.019 | 0.140 | 0.044 | 0.148 | 0.954 | 0.893 | 0.949 | 0.850 |
| 6 doses after removing smallest | LRM | −0.027 | −0.172 | 0.357 | 0.531 | 0.080 | 0.534 | 0.836 | 1.220 | 0.966 | 0.561 | 0.937 | 0.734 |
| | HLRM | 0.004 | 0.027 | −0.010 | −0.005 | 0.028 | 0.160 | 0.089 | 0.125 | 0.434 | 0.247 | 0.384 | 0.253 |
| | BRM | −0.005 | −0.035 | 0.025 | 0.040 | 0.018 | 0.178 | 0.078 | 0.143 | 0.958 | 0.877 | 0.943 | 0.857 |
| 7 doses with less extreme values | LRM | 0.000 | −0.001 | 0.001 | 0.004 | 0.011 | 0.060 | 0.032 | 0.053 | 0.892 | 0.826 | 0.883 | 0.799 |
| | HLRM | 0.000 | 0.000 | 0.001 | 0.002 | 0.011 | 0.061 | 0.032 | 0.056 | 0.665 | 0.597 | 0.657 | 0.564 |
| | BRM | 0.000 | 0.001 | 0.000 | 0.001 | 0.009 | 0.052 | 0.026 | 0.048 | 0.865 | 0.822 | 0.860 | 0.801 |
| 7 doses with extreme values and dose-dependent precision | LRM | −0.052 | −0.215 | 0.524 | 0.462 | 0.130 | 0.622 | 1.123 | 0.984 | 0.965 | 0.521 | 0.921 | 0.726 |
| | HLRM | 0.002 | 0.011 | −0.004 | −0.003 | 0.016 | 0.096 | 0.053 | 0.071 | 0.303 | 0.180 | 0.273 | 0.148 |
| | BRM | 0.000 | 0.012 | −0.001 | −0.005 | 0.009 | 0.142 | 0.054 | 0.080 | 0.957 | 0.851 | 0.932 | 0.798 |
| (c)data simulated using normal error term with SD = 0.05 | | | | | | | | | | | | | |
| 7 doses with extreme values | LRM | 0.079 | −0.463 | 0.560 | 2.399 | 0.393 | 0.924 | 2.128 | 1.853 | 0.948 | 0.612 | 0.942 | 0.591 |
| | HLRM | 0.024 | 0.047 | 0.345 | 1.524 | 0.223 | 1.050 | 1.600 | 2.638 | 0.477 | 0.258 | 0.447 | 0.083 |
| | BRM | 0.013 | 0.029 | 0.010 | 0.095 | 0.129 | 0.524 | 0.371 | 0.377 | 0.857 | 0.861 | 0.869 | 0.855 |

*Appendix 1—table 1 Continued on next page*

*Appendix 1—table 1 Continued*

| Scenario | Method | Bias | | | | RMSE | | | | 95% CI Coverage Probability | | | |
|---|---|---|---|---|---|---|---|---|---|---|---|---|---|
| | | $IC_{50}$ | $IC_{90}$ | $\beta_0$ | $\beta_1$ | $IC_{50}$ | $IC_{90}$ | $\beta_0$ | $\beta_1$ | $IC_{50}$ | $IC_{90}$ | $\beta_0$ | $\beta_1$ |
| 6 doses after removing largest | LRM | 0.079 | −0.338 | 0.337 | 2.194 | 0.276 | 0.749 | 2.054 | 2.239 | 0.968 | 0.779 | 0.975 | 0.675 |
| | HLRM | −0.005 | 0.080 | 0.260 | 0.926 | 0.170 | 0.820 | 1.392 | 2.547 | 0.407 | 0.293 | 0.417 | 0.146 |
| | BRM | 0.017 | 0.001 | 0.007 | 0.160 | 0.124 | 0.512 | 0.387 | 0.430 | 0.879 | 0.835 | 0.912 | 0.812 |
| 6 doses after removing smallest | LRM | 0.022 | −0.403 | 0.666 | 2.232 | 0.335 | 0.965 | 1.936 | 2.220 | 0.972 | 0.610 | 0.898 | 0.681 |
| | HLRM | 0.035 | 0.244 | 0.162 | 0.919 | 0.158 | 0.998 | 1.332 | 2.526 | 0.410 | 0.209 | 0.320 | 0.146 |
| | BRM | 0.009 | −0.021 | 0.030 | 0.160 | 0.119 | 0.511 | 0.358 | 0.420 | 0.874 | 0.754 | 0.862 | 0.819 |
| 7 doses with less extreme values | LRM | 0.005 | −0.077 | 0.086 | 0.362 | 0.084 | 0.383 | 0.693 | 1.215 | 0.905 | 0.795 | 0.895 | 0.813 |
| | HLRM | 0.001 | 0.029 | 0.009 | 0.039 | 0.053 | 0.327 | 0.185 | 0.446 | 0.619 | 0.551 | 0.610 | 0.520 |
| | BRM | 0.002 | −0.001 | 0.012 | 0.062 | 0.054 | 0.296 | 0.187 | 0.311 | 0.861 | 0.810 | 0.858 | 0.815 |
| 7 doses with extreme values and dose-dependent precision | LRM | −0.029 | −0.441 | 0.994 | 1.679 | 0.297 | 0.955 | 1.840 | 1.715 | 0.945 | 0.444 | 0.904 | 0.678 |
| | HLRM | 0.006 | 0.037 | 0.278 | 0.617 | 0.106 | 0.710 | 1.152 | 1.781 | 0.335 | 0.145 | 0.295 | 0.105 |
| | BRM | −0.005 | 0.037 | 0.023 | −0.002 | 0.040 | 0.323 | 0.172 | 0.271 | 0.954 | 0.819 | 0.930 | 0.758 |

**Appendix 1—table 2.** Simulation result of bias, RMSE and 95% CI coverage probability corresponding to beta error terms.

| Scenario | Method | Bias | | | | RMSE | | | | 95% CI Coverage Probability | | | |
|---|---|---|---|---|---|---|---|---|---|---|---|---|---|
| | | $IC_{50}$ | $IC_{90}$ | $\beta_0$ | $\beta_1$ | $IC_{50}$ | $IC_{90}$ | $\beta_0$ | $\beta_1$ | $IC_{50}$ | $IC_{90}$ | $\beta_0$ | $\beta_1$ |
| (a)data simulated using beta error term with $\phi$=35 | | | | | | | | | | | | | |
| 7 doses with extreme values | LRM | 0.017 | −0.304 | 0.142 | 0.653 | 0.163 | 0.481 | 0.658 | 0.686 | 0.924 | 0.697 | 0.909 | 0.566 |
| | HLRM | 0.008 | −0.161 | 0.085 | 0.366 | 0.117 | 0.478 | 0.402 | 0.576 | 0.581 | 0.429 | 0.571 | 0.313 |
| | BRM | 0.003 | −0.014 | 0.018 | 0.073 | 0.074 | 0.339 | 0.219 | 0.275 | 0.835 | 0.818 | 0.832 | 0.829 |
| 6 doses after removing largest | LRM | 0.039 | −0.174 | −0.027 | 0.504 | 0.115 | 0.402 | 0.354 | 0.666 | 0.933 | 0.864 | 0.936 | 0.666 |
| | HLRM | 0.015 | −0.092 | 0.026 | 0.273 | 0.109 | 0.463 | 0.337 | 0.486 | 0.581 | 0.518 | 0.594 | 0.368 |
| | BRM | 0.008 | 0.005 | 0.007 | 0.077 | 0.077 | 0.375 | 0.229 | 0.298 | 0.812 | 0.808 | 0.811 | 0.803 |
| 6 doses after removing smallest | LRM | −0.022 | −0.288 | 0.248 | 0.502 | 0.108 | 0.501 | 0.499 | 0.670 | 0.930 | 0.620 | 0.901 | 0.664 |
| | HLRM | 0.000 | −0.120 | 0.094 | 0.274 | 0.106 | 0.509 | 0.366 | 0.487 | 0.596 | 0.383 | 0.547 | 0.368 |
| | BRM | −0.002 | −0.020 | 0.031 | 0.074 | 0.075 | 0.353 | 0.223 | 0.298 | 0.815 | 0.782 | 0.805 | 0.800 |
| 7 doses with less extreme values | LRM | 0.004 | −0.045 | 0.023 | 0.117 | 0.063 | 0.332 | 0.195 | 0.300 | 0.900 | 0.828 | 0.891 | 0.821 |
| | HLRM | 0.003 | −0.017 | 0.021 | 0.096 | 0.067 | 0.382 | 0.205 | 0.323 | 0.686 | 0.630 | 0.680 | 0.619 |
| | BRM | 0.003 | 0.031 | 0.004 | 0.028 | 0.058 | 0.331 | 0.170 | 0.272 | 0.845 | 0.837 | 0.847 | 0.838 |
| 7 doses with extreme values and dose-dependent precision | LRM | 0.080 | −0.093 | −0.179 | 0.444 | 0.147 | 0.281 | 0.435 | 0.548 | 0.904 | 0.908 | 0.927 | 0.639 |
| | HLRM | 0.061 | −0.200 | −0.063 | 0.753 | 0.199 | 0.553 | 0.676 | 0.967 | 0.484 | 0.447 | 0.520 | 0.231 |
| | BRM | 0.003 | 0.031 | −0.003 | 0.012 | 0.067 | 0.277 | 0.182 | 0.236 | 0.755 | 0.780 | 0.768 | 0.772 |
| (b)data simulated using beta error term with $\phi$=15 | | | | | | | | | | | | | |
| 7 doses with extreme values | LRM | 0.029 | −0.571 | 0.373 | 1.633 | 0.233 | 0.555 | 1.177 | 1.197 | 0.941 | 0.588 | 0.910 | 0.406 |
| | HLRM | 0.016 | −0.334 | 0.225 | 0.983 | 0.172 | 0.657 | 0.725 | 1.186 | 0.561 | 0.353 | 0.544 | 0.213 |
| | BRM | 0.008 | −0.050 | 0.036 | 0.174 | 0.109 | 0.464 | 0.324 | 0.403 | 0.817 | 0.790 | 0.820 | 0.805 |
| 6 doses after removing largest | LRM | 0.074 | −0.360 | −0.043 | 1.269 | 0.164 | 0.516 | 0.635 | 1.192 | 0.942 | 0.823 | 0.939 | 0.556 |
| | HLRM | 0.029 | −0.206 | 0.086 | 0.702 | 0.164 | 0.715 | 0.594 | 0.925 | 0.552 | 0.445 | 0.551 | 0.272 |
| | BRM | 0.018 | −0.028 | 0.019 | 0.203 | 0.117 | 0.527 | 0.355 | 0.446 | 0.778 | 0.783 | 0.794 | 0.776 |
| 6 doses after removing smallest | LRM | −0.042 | −0.564 | 0.620 | 1.280 | 0.157 | 0.596 | 0.890 | 1.180 | 0.939 | 0.503 | 0.905 | 0.555 |
| | HLRM | −0.001 | −0.262 | 0.237 | 0.700 | 0.160 | 0.712 | 0.663 | 0.917 | 0.546 | 0.300 | 0.486 | 0.268 |
| | BRM | −0.001 | −0.071 | 0.066 | 0.192 | 0.115 | 0.499 | 0.345 | 0.449 | 0.787 | 0.733 | 0.782 | 0.777 |

*Appendix 1—table 2 Continued on next page*

*Appendix 1—table 2 Continued*

| Scenario | Method | Bias | | | | RMSE | | | | 95% CI Coverage Probability | | | |
|---|---|---|---|---|---|---|---|---|---|---|---|---|---|
| | | $IC_{50}$ | $IC_{90}$ | $\boldsymbol{\beta}_0$ | $\boldsymbol{\beta}_1$ | $IC_{50}$ | $IC_{90}$ | $\boldsymbol{\beta}_0$ | $\boldsymbol{\beta}_1$ | $IC_{50}$ | $IC_{90}$ | $\boldsymbol{\beta}_0$ | $\boldsymbol{\beta}_1$ |
| | LRM | 0.008 | −0.104 | 0.059 | 0.286 | 0.096 | 0.513 | 0.320 | 0.489 | 0.906 | 0.795 | 0.884 | 0.807 |
| 7 doses with less extreme values | HLRM | 0.009 | −0.030 | 0.048 | 0.240 | 0.103 | 0.726 | 0.325 | 0.517 | 0.694 | 0.602 | 0.678 | 0.598 |
| | BRM | 0.006 | 0.074 | 0.012 | 0.062 | 0.088 | 0.553 | 0.260 | 0.409 | 0.838 | 0.836 | 0.841 | 0.847 |
| 7 doses with extreme values and dose-dependent precision | LRM | 0.165 | −0.202 | −0.425 | 1.114 | 0.218 | 0.374 | 0.771 | 0.959 | 0.916 | 0.886 | 0.941 | 0.534 |
| | HLRM | 0.141 | −0.347 | −0.250 | 2.055 | 0.291 | 1.027 | 1.270 | 1.798 | 0.515 | 0.440 | 0.576 | 0.136 |
| | BRM | 0.010 | 0.053 | −0.008 | 0.050 | 0.104 | 0.423 | 0.281 | 0.368 | 0.763 | 0.790 | 0.764 | 0.769 |
| (c)data simulated using beta error term with $\phi=5$ | | | | | | | | | | | | | |
| | LRM | 0.042 | −0.851 | 0.748 | 3.414 | 0.270 | 0.527 | 1.611 | 1.421 | 0.948 | 0.423 | 0.898 | 0.198 |
| 7 doses with extreme values | HLRM | 0.040 | −0.629 | 0.689 | 3.147 | 0.278 | 1.518 | 1.480 | 2.112 | 0.681 | 0.241 | 0.644 | 0.094 |
| | BRM | 0.008 | −0.011 | 0.047 | 0.191 | 0.161 | 0.606 | 0.589 | 0.767 | 0.811 | 0.800 | 0.815 | 0.811 |
| | LRM | 0.106 | −0.636 | 0.181 | 2.962 | 0.234 | 0.865 | 1.401 | 1.604 | 0.949 | 0.684 | 0.938 | 0.356 |
| 6 doses after removing largest | HRLM | 0.075 | −0.098 | 0.276 | 2.410 | 0.274 | 8.599 | 1.351 | 2.082 | 0.607 | 0.362 | 0.600 | 0.142 |
| | BRM | 0.007 | −0.115 | 0.112 | 0.355 | 0.166 | 0.643 | 0.621 | 0.761 | 0.762 | 0.752 | 0.773 | 0.769 |
| | LRM | −0.039 | −0.871 | 1.126 | 2.935 | 0.231 | 0.649 | 1.428 | 1.620 | 0.953 | 0.380 | 0.883 | 0.362 |
| 6 doses after removing smallest | HLRM | 0.004 | −0.371 | 0.773 | 2.379 | 0.261 | 5.238 | 1.464 | 2.068 | 0.599 | 0.206 | 0.492 | 0.149 |
| | BRM | 0.008 | −0.087 | 0.070 | 0.328 | 0.166 | 0.588 | 0.509 | 0.724 | 0.766 | 0.755 | 0.772 | 0.773 |
| | LRM | 0.008 | −0.456 | 0.275 | 1.126 | 0.152 | 0.515 | 0.679 | 1.041 | 0.915 | 0.639 | 0.874 | 0.707 |
| 7 doses with less extreme values | HLRM | 0.007 | −0.315 | 0.236 | 0.923 | 0.164 | 0.767 | 0.651 | 1.047 | 0.681 | 0.475 | 0.644 | 0.491 |
| | BRM | 0.002 | −0.088 | 0.093 | 0.318 | 0.132 | 0.588 | 0.423 | 0.660 | 0.839 | 0.812 | 0.834 | 0.835 |
| 7 doses with extreme values and dose-dependent precision | LRM | 0.245 | −0.378 | −0.646 | 2.300 | 0.274 | 0.470 | 1.207 | 1.262 | 0.908 | 0.833 | 0.947 | 0.343 |
| | HLRM | 0.209 | −0.578 | −0.233 | 4.982 | 0.332 | 6.852 | 2.016 | 2.017 | 0.683 | 0.381 | 0.728 | 0.050 |
| | BRM | 0.025 | 0.171 | −0.032 | 0.074 | 0.161 | 0.762 | 0.426 | 0.549 | 0.757 | 0.796 | 0.761 | 0.770 |

**Appendix 1—table 3.** The first example of REAP application with B-cell lymphoma data, corresponding with *Figure 5*.

$IC_{50}$ estimations are ranked from low to high. Hypothesis testings on equal potency (i.e., concentration for $IC_{50}$) were conducted pairwise with the group right above (one rank lower). Jeko-1 has the highest potency and the difference of $IC_{50}$ estimations between Jeko-1 and Jeko-R is significant with a *P*-value <0.0001. The B-cell lymphoma dataset is available on Github (*Fang et al., 2022*).

| Model | Intercept | Slope (m) | Std. Err for m | P-value for m>1 | $IC_{50}$ estimation | Std. Err for $IC_{50}$ estimation | Pairwise comparison |
|---|---|---|---|---|---|---|---|
| Jeko-1 | −4.807 | −1.252 | 0.155 | 0.0519 | 0.021 | 0.008 | - |
| Jeko-R | −4.305 | −1.822 | 0.112 | <.0001 | 0.094 | 0.006 | <0.0001 |
| Rec-1 | −5.304 | −2.63 | 0.091 | <.0001 | 0.133 | 0.004 | <0.0001 |
| Mino | −2.684 | −1.474 | 0.141 | 0.0004 | 0.162 | 0.015 | 0.0656 |
| Jeko-NO #1 | −2.012 | −1.192 | 0.135 | 0.0769 | 0.185 | 0.021 | 0.3755 |
| MAVER-1 | −2.21 | −1.37 | 0.125 | 0.0015 | 0.199 | 0.021 | 0.6312 |
| Jeko-NO #11 | −1.459 | −1.267 | 0.152 | 0.0398 | 0.316 | 0.038 | 0.0114 |
| JVM2 | −1.056 | −1.271 | 0.135 | 0.0223 | 0.436 | 0.055 | 0.0818 |

**Appendix 1—table 4.** The output for the estimated dose-response curve of anti-viral drugs under the same biological batch with SARS-CoV-2 data.

Calpain inhibitor IV has the highest potency (*P*-value = 0.0038). The reconstructed SARS-CoV-2 dataset is available on Github (*Fang et al., 2022*).

| Model | Intercept | Slope (m) | Std. Err for m | P-value for m>1 | $EC_{50}$ estimation | Std. Err for $EC_{50}$ estimation | Pairwise comparison |
|---|---|---|---|---|---|---|---|
| CalpainInhibitorIV | 0.678 | 0.725 | 0.114 | 0.9918 | 0.393 | 0.103 | - |
| Chloroquine | −1.013 | 0.84 | 0.135 | 0.8813 | 3.337 | 0.88 | 0.0038 |
| Remdesivir | −1.791 | 0.797 | 0.12 | 0.9553 | 9.469 | 2.638 | 0.0282 |
| Hydroxychloroquine | −1.485 | 0.562 | 0.075 | 1 | 14.074 | 4.994 | 0.4445 |
| E64d (Aloxistatin) | −3.211 | 0.861 | 0.129 | 0.8587 | 41.61 | 15.473 | 0.1242 |

