## [Editor Report]

This article proposes methodology and accompanying software for robustly fitting dose-response curves where response is a number between 0 and 1. When response is transformed using the common logistic transformation, values close to 0 or 1 become large in magnitude, unduly influencing the fitted curve after back-transformation and introducing bias in the estimate of certain parameters. As demonstrated through simulation and application to real data, the proposed approach, called Robust and Efficient Assessment of Potency, is less perturbed by these extreme measurements.

---

## [Decision Letter]

**Decision letter after peer review:**

Thank you for submitting your article "Robust and Efficient Assessment of Potency (REAP): A Quantitative Tool for Dose-response Curve Estimation" for consideration by *eLife*. Your article has been reviewed by 2 peer reviewers, including Philip Boonstra as the Reviewing Editor and Reviewer #1, and the evaluation has been overseen by and Aleksandra Walczak as the Senior Editor.

Essential revisions:

1) Following Reviewer 2's first comment, please clearly state what LRM is.

2) Related to this and as suggested above, please consider also comparing the REAP approach to a version of LRM with a heavy tailed error distribution such as t-distribution with 3 degrees of freedom, which would also seem to possess the same robustness properties as REAP.

3) Please conduct additional stress testing and debugging of the shiny app, if it is intended to be included and advertised in the manuscript. See Reviewer 1's comments for specific suggestions.

*Reviewer #1 (Recommendations for the authors):*

I would suggest running additional stress tests on the web app to address some of the bugs above. You might also add some additional features. For example, there is a little help key next to 'Add effect estimation' that, when you hover over it, explains what that does. Could more such keys be added to other user inputs?

As to the methodology, it seems to me like it would be worthwhile to consider an approach such as what I describe in my public review – essentially the linear model but with a heavy tailed error distribution that will be less sensitive to extreme values.

*Reviewer #2 (Recommendations for the authors):*

The topic of this manuscript is very interesting to many researchers who work on drug screening tests and development. The manuscript is well developed and well written. How I do have some comments for the authors to address.

(1) The authors should clearly state what the LRM is, I assume it is model (3) based on normal error assumption. However, the authors should clearly spell it out.

(2) The parameter β is estimated by minimizing (8) or (9), are the two equations (8) and (9) equivalent? If so, the authors should make the connection clear.

(3) In the method session, the authors mentioned the stability condition for the selection of the tuning parameter. What is the stability condition? The authors stated that "If the stability condition is satisfied before *α*max is reached, then optimal *α* equals the minimal value in the grid of *α*k.". Do the authors imply that the optimal *α* equals the minimal value in the grids which met the stability condition?

Overall, this is a nice manuscript and will make a nice contribution to the field involving in-vitro drug screening and testing.

---

## [Author Response]

Essential revisions:Reviewer #1 (Recommendations for the authors):I would suggest running additional stress tests on the web app to address some of the bugs above. You might also add some additional features. For example, there is a little help key next to 'Add effect estimation' that, when you hover over it, explains what that does. Could more such keys be added to other user inputs?

We thank the reviewer for the instructive suggestion and conducted multiple experiments in different data patterns to stress test the web app. We have fixed the following bugs and added instructions for some features:

1. Solve the triplicate model assessments and comparison when there is only one agent in the dataset

2. Add a hover box next to “Model Comparisons” to indicate how to select the two check boxes for effect, slope and model comparisons

3. Add a hover box next to “Width of error bar” indicating the range of the error bar width should be within (0, 0.1). The input width is restricted within the range. If the input is outside of (0, 0.1), an error message will appear below the input box and the input value will be capped at 0.1.

4. Solve the problem when there are negative dose levels in data.

5. Solve the problem of missing sample mean and error bar in dose-response curve plotting given either out of range or missing dose responses.

As to the methodology, it seems to me like it would be worthwhile to consider an approach such as what I describe in my public review – essentially the linear model but with a heavy tailed error distribution that will be less sensitive to extreme values.

This is a great suggestion and we have added the simulation under both normal error term and β error term to compare the robust β regression framework with standard linear regression model with normal error distribution and the heavy-tailed linear model with t (df = 3) error distribution. In general, the heavy-tailed linear regression model demonstrates great potential to improve point estimation, but the coverage probabilities of its nominal 95% confidence interval are consistently underestimated below the level of 50% if with extreme values. The comparison results have been updated in the simulation Results section, Figure 4, Supplementary Figure 1 and Supplementary Tables 1 and 2.

Reviewer #2 (Recommendations for the authors):The topic of this manuscript is very interesting to many researchers who work on drug screening tests and development. The manuscript is well developed and well written. How I do have some comments for the authors to address.(1) The authors should clearly state what the LRM is, I assume it is model (3) based on normal error assumption. However, the authors should clearly spell it out.

We thank the reviewer for the instructive suggestion. LRM has been spelled out in the Methods section “A linear regression model (LRM) can be applied in the form of equation (3) with a standard normal distribution error.” Additionally, we also include the specifications on the heavy-tailed linear regression model (HLRM) and on the robust β-regression model (BRM) in the Methods section and provide the linkage in figure captions.

(2) The parameter β is estimated by minimizing (8) or (9), are the two equations (8) and (9) equivalent? If so, the authors should make the connection clear.

We apologize for the confusion. Equation (9) is simplified from equation (8), with technical details in Section 2.2 (Pg. 875) of Ghosh (2019). In the Methods section of the manuscript, additional descriptions are added:

- In equation (8),θ=(β,ϕ)T

- After mathematically simplifying equation (8), *θ* can be equivalently estimated by minimizing the objective function using the estimation equations (9).

(3) In the method session, the authors mentioned the stability condition for the selection of the tuning parameter. What is the stability condition? The authors stated that "If the stability condition is satisfied before αmax is reached, then optimal α equals the minimal value in the grid of αk.". Do the authors imply that the optimal α equals the minimal value in the grids which met the stability condition?

We thank the reviewer for the question for us to clarify the estimation methods. The stability condition is SQVαk<L. The default choice of L is 0.02. The optimal *α* is chosen in different situations:

1. If the stability condition is satisfied before α*_max_* is reached, then optimal *α* equals the minimal value in the grid of αk.

2. If α*_max_* is reached, then optimal *α* equals 0, which is equivalent to the maximum likelihood estimation.

The above descriptions are also illustrated in the paragraph after the definition of SQV in the Methods section.